# Endometritis Changes the Neurochemical Characteristics of the Caudal Mesenteric Ganglion Neurons Supplying the Gilt Uterus

**DOI:** 10.3390/ani10050891

**Published:** 2020-05-20

**Authors:** Barbara Jana, Jarosław Całka

**Affiliations:** 1Division of Reproductive Biology, Institute of Animal Reproduction and Food Research of the Polish Academy of Sciences, Tuwima 10, 10-748 Olsztyn, Poland; 2Department of Clinical Physiology, Faculty of Veterinary Medicine, University of Warmia and Mazury, Oczapowskiego 14, 11-041 Olsztyn, Poland; calkaj@uwm.edu.pl

**Keywords:** uterus, inflammation, sympathetic innervation, caudal mesenteric ganglion, chemical plasticity, gilt

## Abstract

**Simple Summary:**

Uterine inflammation is a very frequent pathology in domestic animals leading to disturbances in reproductive processes and causing significant economic losses. The uterus possesses nerves from either the autonomic or sensory part of the peripheral nervous system. Most of the uterus-projecting neurons are localized in the caudal mesenteric ganglion. These neurons synthesize and release numerous biologically active substances in the uterus, which regulate uterine functions. The effect of inflammation on uterine innervation is poorly recognized. This study showed that *Escherichia coli*-induced uterine inflammation in pig led to a reduction in the total population of uterine neurons in the caudal mesenteric ganglion, and in the populations of these cells in the dorsal and central areas of this ganglion. In the caudal mesenteric ganglion of gilts after intrauterine bacterial injection, the population of uterine neurons presenting positive staining for dopamine-β-hydroxylase (an enzyme participating in noradrenaline synthesis) and negative staining for galanin, as well as the population of uterine neurons presenting negative staining for dopamine-β-hydroxylase but positive staining for neuropeptide Y, were decreased. In these gilts, there were increased numbers of uterine neurons which, besides dopamine-β-hydroxylase, also expressed neuropeptide Y, galanin and vasoactive intestinal peptide. The above changes suggest that inflammation of the gilt uterus may affect the function(s) of this organ by its action on the neurons of the caudal mesenteric ganglion.

**Abstract:**

This study analyzed the influence of uterine inflammation on the neurochemical characteristics of the gilt caudal mesenteric ganglion (CaMG) uterus-supplying neurons. The horns of uteri were injected with retrograde tracer Fast Blue on day 17 of the first studied estrous cycle. Twenty-eight days later (the expected day 3 of the third studied estrous cycle), either saline or *Escherichia coli* suspension were administered into each uterine horn. Only the laparotomy was done in the control gilts. After 8 days, the CaMGs and uteri were harvested. The infected gilts presented a severe acute endometritis. In the CaMGs, the populations of uterine perikarya possessing dopamine-β-hydroxylase (DβH) and/or neuropeptide Y (NPY), somatostatin (SOM), galanin (GAL) and vasoactive intestinal polypeptide (VIP) were analyzed using the double immunofluorescence method. In the CaMG, bacterial injection decreased the total number of the perikarya (Fast Blue-positive), the small and large perikarya populations in the dorsal and central regions, and the small and large perikarya populations coded DβH+/GAL- and DβH-/NPY+. After bacterial treatment, there was an increase in the numbers of small and large perikarya coded DβH+/NPY+, small perikarya coded DβH+/GAL+ and DβH+/SOM- and large perikarya coded DβH+/VIP+. To summarize, uterine inflammation influences the neurochemical characteristics of the CaMG uterus-supplying neurons, which may be important for pathologically changed organ functions.

## 1. Introduction

Uterine inflammatory process is a very common condition in domestic animals, which leads to economic losses. Although this disorder may occur in females which have not yet given birth (after natural mating, insemination), uterine inflammation takes place predominantly after parturition as a result of disturbances in uterine involution and/or immunological reaction. Metritis/endometritis is evoked for the most part by bacteria [1,2,3,4]. Situations such as difficult labor, fetal membrane retention and uterine contractility disturbances are conducive to the origin of this pathology. Mild uterine inflammation does not cause any serious deviations in the estrous cycle course. In turn, in more advanced cases of inflammation, an inflammatory exudate (mucopurulent) is present in the uterine cavity. The myometrium of animals, especially cows, loses its ability to contract [5]. During uterine inflammation, the production and release of cytokines [6,7,8,9], prostaglandin (PG)F_2α_, PGE_2_ [2,10], PGI_2_ [11], leukotrienes (LT)B_4_ and LTC_4_ [12], as well as nitric oxide [13] are markedly increased. 

The uterus in pigs is innervated by nerves from either the autonomic or sensory part of the peripheral nervous system. In the pig, most of the uterus-projecting neurons (further referred to as uterine neurons or uterine perikarya), indicated by fluorescent retrograde neuronal tracer Fast Blue (FB), exist in the caudal mesenteric ganglion (CaMG), where they form the “uterine region”. The vast majority of all uterine perikarya in the CaMG are noradrenergic. These structures possess enzymes that synthetize noradrenaline (NA): tyrosine hydroxylase (TH) and dopamine-β-hydroxylase (DβH). However, other neurotransmitters such as galanin (GAL), neuropeptide Y (NPY), vasoactive intestinal polypeptide (VIP) and somatostatin (SOM) are also expressed in those perikarya. Moreover, in the sow, these neurotransmitters are present in the nerve fibers in the area of CaMG [14,15] and in the uterus, under luminal epithelium, near the blood vessels, endometrial glands and myocytes of the myometrium [16,17]. It is known that NA, acting through α- and/or β-adrenergic receptors, controls the contractile activity of the uterus [18,19] and PGs production in endometrial and myometrial cells under physiological conditions [20,21]. NPY, via its receptor subtype 1, stimulates the contractility of myometrium [22] and uterine arteries [23,24]. Similarly, GAL [25] and VIP [26] affect uterine contractility, while SOM regulates endometrial cell proliferation and motility [27]. 

Data are scarce on the relationship between uterine inflammation and morphology, as well as on the immunochemical characteristics of neurons projecting to the uterus. In rats, the inflammation of the uterus evoked abnormal behavior associated with visceral pain [28] and upregulated the uterine substance P (SP)-positive perikarya numbers in the dorsal root ganglia (DRGs) [29]. Due to its embryological, anatomical and physiological similarities to humans, the pig is a valuable species in biomedical research, also for the study of uterine function [30,31]. One published study conducted on pigs, describing the influence of uterine inflammation on the innervation of the uterus, indicated that in gilts, *Escherichia coli* (*E. coli*)-induced inflammatory state changes the morphological and neurochemical characteristics of the DRG neurons supplying the uterus [32]. In addition, it was found that there was a reduction in the total population of nerve fibers in the inflamed porcine uterus, including DβH-positive around inflamed uterine structures [33]. Based on these findings, it is hypothesized that the uterine inflammatory state affects the neurochemical characteristics of the uterine neurons in CaMG. To test this hypothesis, the effect of uterine inflammation was examined on (1) the total number of uterine perikarya and their size and localization, and on (2) the uterine perikaryal cell count containing DβH and/or NPY, SOM, GAL or VIP in the gilt CaMG. 

## 2. Materials and Methods 

### 2.1. Animals 

The experiment was performed taking into account the principles of animal care (National Institute of Health publication No. 86–23, revised in 1985) and the specifics of national law concerning animal protection. The Local Ethics Committee of the University of Warmia and Mazury in Olsztyn approved all procedures and granted consent (no. 65/2015). 

This research was carried out on 11 gilts (crossbred Large White × Landrace) at the age of 7–8 months and body weight (BW) of approximately 90–120 kg. By using a tester boar, behavioral estrus was estimated. In all studied gilts, disruption in reproductive state did not occur, vaginal discharges were not revealed, and the second estrous cycle was regular. The gilts were kept in laboratory conditions (Faculty of Veterinary Medicine, University of Warmia and Mazury, Olsztyn, Poland). They were maintained in individual pens with an area of about 5 m^2^ under the following conditions: natural daylight—14.5 ± 1.5 h, night—9.5 ± 1.5 h and temperature 18 ± 2 °C. The pigs were fed with commercial diets according to their nutritional requirements and had access to water. After transport, the pigs were divided (randomly) into three groups: *Escherichia coli* (*E. coli* group, n = 4), the saline (SAL group, n = 3)-treated gilts and control (CON control, n = 4) gilts—subjected to “sham” operation (details are below). The study was started after three days (adaptive period). During the experiment, the animals were not medically treated. 

### 2.2. Experimental Procedures

The experimental procedure was previously described [32]. On day 17 of the first studied estrous cycle (day 0 of the study), before surgery, the gilts were pre-medicated with atropine (0.05 mg/kg, administered intramuscularly (i.m.); Atropinum sulf. WZF, Warszawskie Zakłady Farmaceutyczne Polfa S.A., Warsaw, Poland), azaperone (2 mg/kg BW, administered i.m. Stresnil, Janssen Pharmaceutica, Beerse, Belgium) and ketamine hydrochloride (10 mg/kg BW, administered intravenously (i.v.); Ketamina, Biowet, Puławy, Poland). General anesthesia was reached with ketamine hydrochloride and prolonged by the application of supplementary doses of this medicine (1 mg/kg BW every 5 min, administered i.v.). After laparotomy, the uterine horns were injected with Fast Blue (FB, 5% aqua solution, EMS-CHEMIE, GmbH, Gross-Umstadt, Germany) to indicate the cell bodies of neurons projecting to the uterus. FB was administered using a Hamilton syringe with a 26-gauge needle into the wall of each uterine horn in paracervical, middle and paraoviductal portions. In each part (ring about 2 cm wide), 13 FB injections were done (volume of each injection—2 μL, total volume per place—26 μL). The needle of the Hamilton syringe was kept in each place for 1 min following injection to limit the leakage of FB outside the uterine tissue. Next, the place of injection was rinsed using isotonic saline and wiped with gauze. 

Twenty-eight days later (the necessary period for FB to achieve the external sources of innervation of the uterus in pigs), on the expected day 3 of the third studied estrous cycle, the gilts were anaesthetized (as explained above). In gilts, after laparotomy was done, either 50 mL of *E. coli* suspension (*E. coli* group; 1 mL of suspension containing 10^9^ colony-forming units, strain O25:K23/a/:H1; National Veterinary Research Institute, Department of Microbiology, Puławy, Poland), or 50 mL of saline solution (SAL group) were administered into both uterine horns. In the gilts of the CON group, only laparotomy was carried out. After 8 days (the expected day 11 of the third studied estrous cycle), euthanasia of gilts was performed using an overdose of ketamine hydrochloride (administered i.v.) and the gilts were transcardially perfused via the ascending aorta with 4% buffered paraformaldehyde (pH 7.4). Next, the bilateral CaMGs were obtained from gilts of all groups. The ganglia were post-fixed by immersion in the same fixative for 10 min, then washed with 0.1 M PB (pH 7.4) for two days and stored at 4 °C in an 18% buffered sucrose solution (pH 7.4), with natrium azide (0.001%). Later, the CaMGs were kept at −80 °C until further examination. For the microscopic study, the fragments of uterine horns were fixed in 4% paraformaldehyde solution (pH 7.4) for 24 h, and the tissues were then washed in 0.1 M phosphate-buffered saline (PBS, pH 7.4) and embedded in paraffin. The findings of the histological assessment of uteri were published previously [32]. 

### 2.3. Immunohistochemical Analysis 

Serial cryostat sections (with a thickness of 10 μm, Frigocut, Reichert-Jung, Nussloch, Germany) of the CaMG were placed on chrome alum-coated slides. The presence of FB-positive perikarya was checked in serial sections of the bilateral CaMGs using an Olympus BX51 microscope (Olympus, Warsaw, Poland), equipped for epi-illumination fluorescence microscopy (V1 module: excitation filter 330−385 nm, barrier filter 420 nm). Sections with FB-positive perikarya were used for double-labeling immunofluorescence to determine DβH and/or NPY, SOM, GAL and VIP immunoreactivity [34]. The sections were dried (at 32 °C, 45 min), rinsed in a phosphate buffer with 0.8% sodium chloride and 0.02% potassium chloride (PBS, 3 × 10 min) and incubated in 10% normal goat serum in PBS with 0.3% Triton X-100 (Sigma, Saint Louis, MO, USA) and 1% bovine serum albumin (BSA; Sigma, USA) for 20 min. Next, the sections were incubated overnight (at 4 °C) with primary antibodies diluted in PBS containing 0.3% Triton X-100 and 1% BSA, raised against DβH (rabbit polyclonal, Cat. # AB1585, Merck Millipore, Kenilworth, NJ, USA, 1:500) and/or NPY (mouse monoclonal, Cat. # ABS 028-08-02, ThermoFisher Scientific, Waltham, MA, USA, 1:1000), SOM (rat monoclonal, Cat # 8330-0009, AbD Serotec, Kidlington, UK, 1:60), GAL (guinea pig polyclonal, Cat. # T-50-36, Penisula, San Carlos, CA, USA, 1:800) or VIP (mouse polyclonal, Cat # 9535-0504, BioGene Ltd., Huntingdon, UK, 1:2000). On the next day, the sections were rinsed (PBS, 5 × 15 min) and incubated with secondary antibodies (in PBS with 0.25% BSA and 0.1% Triton X-100) for 4 h (Alexa Fluor 488 nm donkey anti-rabbit, Cat # A21206, Alexa Fluor 546 nm donkey anti-mouse, Cat # A10036, Alexa Fluor 546 nm donkey anti-rat, Cat # A11081 and Alexa Fluor 546 nm donkey anti-guinea pig, Cat # A11074, all from ThermoFisher Scientific, Waltham, MA, USA, and diluted 1:1000) to visualize the following antibody combinations: DβH/NPY, DβH/SOM, DβH/GAL and DβH/VIP. After that, the sections were rinsed (PBS, 3 × 5 min) and covered with a polyethylene glycol/glycerin solution with 1,4-diazabicyclo[2.2.2]octane (DABCO, Sigma, USA). To control for immunofluorescence specificity, standard tests (pre-absorption for the applied antisera with the respective antigen at a content of 20–50 μg antigen/mL diluted antiserum, exclusion of primary or secondary antisera and replacement by non-immune sera of all the primary antisera used) were conducted. The specificity of retrograde tracing was examined by the use of the various tests presented previously [32]. FB-positive and double-immunostained perikarya were investigated and photographed with the appropriate filter sets for fluorescein isothiocyanate (FITC, B1 module, excitation filter 450–480 nm, barrier filter 515 nm) and CY3 (G1 module excitation filter 510–550 nm, barrier filter 590 nm). DβH-, NPY-, SOM-, GAL-, VIP-immunoreactive and/or all FB-positive perikarya were calculated in every fourth section of the bilateral CaMGs. CaMG uterine perikarya profiles with a visible nucleus were only scored to avoid double counting. The perikarya distribution was determined for particular regions of CaMG, taking into account the method reported previously [34]. In brief, the cranial region of the CaMG constitutes an area corresponding to I splanchnic lumbar nerve and intermesenteric nerve, the dorsal region forms the area corresponding to II and III splanchnic nerves, the caudal region composes the area corresponding to IV splanchnic lumbar nerve and hypogastric nerve and the ventral region constitutes the area corresponding to caudal colonic nerves. In addition, the perikarya were classified as small (with a diameter of 23 ± 10 μm) or large (with a diameter of 51 ± 17 μm) [34], using Cell F Imaging Software (Olympus, PL). The perikarya sizes were estimated by measurements of their long and short axis. The average value of the perikaryon diameter was assigned by the use of the equation d = √l × k, where d equals the diameter of a circle with a surface area which is the most similar to the surface area of an ellipsoidal figure with a long axis (l) and a short axis (k) [35]. By adding the small and large perikaryal cell count from all areas of the CaMG, the total population of perikarya for each studied group was determined. The images were captured by a digital camera connected to a PC and analyzed with AnalySIS software (version 3.02, Soft Imaging System, Münster, Germany).

### 2.4. Statistical Analysis

Data obtained from the CaMGs of gilts from the CON, SAL and *E. coli* groups were averaged per one CaMG, region, small perikarya, large perikarya and perikarya, with individual neurochemical characteristics for each group. Data concerning neurochemical characteristics were expressed as percentages of the total population of small or large uterine perikarya stained for two substances in each group, accepted as 100%. Statistical significances (mean ± standard error of the mean (SEM)) were estimated by a one-way analysis of variance (ANOVA) followed by the Bonferroni test (Statistica 13 software, StatSoft Inc., Tulsa, OK, USA). Differences were evaluated as significant if the probability was *p* < 0.05. Before the experiment, a statistical power calculation was not performed. The number of gilts in the studied group was based on the earlier studies, in which 3 to 4 animals were used for neuro-immunofluorescence experiments. 

## 3. Results

### 3.1. The Population and Localization of the CaMG Uterine Perikarya 

In the *E. coli* group, the total number of FB-positive uterine perikarya (small and large perikarya combined) was lower (*p* < 0.01) compared to the CON and SAL groups (70 ± 24.2 versus 228 ± 45.3, 245 ± 35.1, respectively). In the *E. coli* group, a reduction (*p* < 0.01) in the number of small perikarya was found in relation to the CON and SAL groups (45.5 ± 14.3 versus 137.9 ± 18.4, 146 ± 21.4, respectively). In the gilts, after intrauterine infusion of bacteria, the number of large perikarya was also lower (*p* < 0.05) than in the CON and SAL groups (30.8 ± 6.5 versus 95 ± 25.1, 102 ± 16.8, respectively). 

In relation to the CON and SAL groups, in the *E. coli* group, the numbers of small perikarya was lower in the cranial (*p* < 0.01), caudal (*p* < 0.001), dorsal (CON—*p* < 0.05, SAL—*p* < 0.01) and central (*p* < 0.01) ganglional areas (Figure 1A). The populations of large perikarya in the dorsal (*p* < 0.05), ventral (*p* < 0.01) and central (*p* < 0.01) CaMG areas of the bacterial-treated gilts were reduced compared to other groups (Figure 1B). 

### 3.2. The Populations of the CaMG Uterine Perikarya Expressing DβH, NPY, SOM, GAL and VIP 

In the *E. coli* group, the numbers of small (CON—*p* < 0.05, SAL—*p* < 0.001) and large (*p* < 0.001) perikarya presenting positive staining for DβH and NPY increased, while the numbers of these perikarya presenting negative staining for DβH and positive staining for NPY were reduced (*p* < 0.05) in relation to other groups. The population of large perikarya presenting negative staining for DβH and NPY after intrauterine bacterial infusion was also lower (*p* < 0.05) than in the CON group (Figure 2A, Figure 3A–H). The number of small perikarya expressing DβH but not SOM increased in the *E. coli* group in relation to the CON (*p* < 0.01) and SAL (*p* < 0.001) groups. The numbers of small and large uterine perikarya presenting negative staining for DβH and SOM were reduced compared to the CON (*p* < 0.01) and SAL (*p* < 0.001) groups (Figure 2B, Figure 3I–P). Intrauterine injections of *E. coli* increased (*p* < 0.001) the number of small perikarya expressing DβH and GAL compared to other groups. In turn, in the *E. coli* group, the numbers of small (CON—*p* < 0.05, SAL—*p* < 0.01) and large (*p* < 0.05) perikarya presenting positive staining for DβH and negative for GAL and the number of small perikarya with negative staining for DβH and GAL (*p* < 0.05) were reduced compared to other groups (Figure 2C, Figure 4A–H). The population of large perikarya presenting positive staining for DβH and VIP was higher after intrauterine bacterial treatment than in the CON (*p* < 0.05) and SAL (*p* < 0.01) groups. The small (CON—*p* < 0.001, SAL—*p* < 0.01) and large (CON—*p* < 0.01, SAL—*p* < 0.001) perikarya populations without positive staining for DβH and VIP were reduced compared to other groups (Figure 2D, Figure 4I–P). The numbers of uterine perikarya expressing DβH and/or NPY, SOM, GAL or VIP as well as those without these substances within the gilt CaMG from the CON, SAL and *E. coli* groups are given in Figure 2A–D.

## 4. Discussion

This study, for the first time, showed how the inflammation of the uterus alters the neurochemical characteristics of the CaMG’s uterine perikarya. The results of the histopathological assessment of pig uteri used in the current experiment were presented previously. In the *E. coli*-treated uteri, severe acute endometritis was revealed [32]. In the CaMG of gilts from the CON and SAL groups, the total number, size and localization of uterine perikarya, and the numbers of these structures expressing DβH and/or NPY, SOM, GAL or VIP were similar. This shows that administration of saline into the uterus did not significantly change the neurochemical features of the examined perikarya. 

The current study revealed that after intrauterine infusion of bacteria in the gilt, the total population of uterine perikarya in CaMG was reduced. Uterine inflammation also led to a decrease in the total population of uterine perikarya in the DRGs of animals examined in the current experiment [32]. Although in the present study the steroid levels in peripheral blood of gilts were not determined, it is possible that they may influence the uterine perikarya populations in the *E. coli* group. In pigs and cows with an inflamed uterus, the level of 17β-estradiol (E_2_) was decreased, and in pigs, the androstenedione level was also augmented [10,36,37,38]. A drop in the total number of CaMG uterine perikarya in pigs with uterine inflammation may be a consequence of a decreased level of E_2_ and an elevated level of androgens. The neuroprotective properties of estrogen are connected with their ability to activate various membrane-associated intracellular signaling pathways and nuclear receptors for estrogen (ERs) and to induce growth factors’ production, as well as their antioxidant actions [39,40]. Moreover, ERs are expressed in rat uterine perikarya in the DRGs [41] and porcine ovary perikarya in the CaMG [42]. The drop in CaMG of ovarian neurons and an augmentation in the set of these cells with androgen receptors (ARs) after long-term testosterone (T) administration in gilts was reported [43]. In addition, androgens have the ability to decrease growth factor production in different tissues [44,45]. It is possible that the reduced survival of uterine perikarya in the CaMG revealed in the present study may also be a consequence of increased synthesis and the release of inflammatory factors in the course of uterine inflammation. This is supported by reports that show elevated levels of pro-inflammatory cytokines (tumor necrosis factor α (TNF-α), interleukin 1 β), as well as PGF_2α_, LTB_4_ and LTC_4_ in peripheral blood of females with an inflamed uterus [7,9,10,12] and the presence of TNF-α, LTB_4_ and LTC_4_ receptors in the perikarya within DRG [46,47,48]. It is known that pro-inflammatory cytokines participate in neuron death by toxic free radical generation [49]. The loss of both small and large perikarya in the CaMG after intrauterine *E. coli* administration may be associated with a similar content of the ERs and/or ARs [42,43], as well as receptors for inflammatory mediators [46,47,48]. However, the exact mechanism of steroid hormone- and inflammatory factor-dependent drop in the neuronal CaMG number needs further studies.

The current study found a decrease in the gilt CaMG populations of small and large uterine perikarya following intrauterine injection of bacteria. In addition, in the DRGs of gilts used in the current study, uterine inflammation led to a drop in the population of small uterine perikarya [32]. Although in the present experiment the effect of the inflammatory process on the dynamic changes in perikarya size has not been determined, the revealed changes in the numbers of small and large uterine perikarya may result from their atrophy or hypertrophy, although this hypothesis needs confirmation. The small and large uterine perikarya were found in all examined areas of the CaMG of gilts from the CON, SAL and *E. coli* groups. In the CaMG of gilts after intrauterine bacterial injection, the numbers of small and large perikarya were reduced in the dorsal and central regions. In addition, uterine inflammation decreased the numbers of small perikarya in the cranial and caudal regions and the number of large perikarya in the ventral region. Previously, it was mentioned that the changes in populations of uterine perikarya inside the particular DRGs of gilts with uterine inflammation were not related to distinct regions [32]. Partly compatible to the current findings is another study, which reported a drop in ovarian perikarya (small, large) in particular areas of the CaMG of E_2_-injected gilts [42] and a reduction of small ovarian perikarya in these ganglia of T-injected gilts [43]. 

In the CaMG of gilts after intrauterine bacterial treatment, a rise was found in the number of small and large uterine perikarya expressing DβH and NPY, the number of small uterine perikarya presenting positive staining for DβH and GAL, the number of small uterine perikarya immunoreactive to DβH but immunonegative to SOM and the number of large uterine perikarya with positive staining for DβH and VIP. In contrast, a decrease was noted in the number of small and large uterine perikarya expressing DβH but not GAL, in the number of small and large uterine perikarya immunonegative to DβH, but immunoreactive to NPY, as well as in the number of uterine perikarya presenting negative staining for the studied substances (except for small uterine perikarya immunonegative to DβH and NPY and large uterine perikarya immunonegative to DβH and GAL). Similarly, expression of GAL was increased in the uterine perikarya in the Th10-S4 DRGs of gilts with uterine inflammation used in the present study [32] and in the colon-projecting neurons in the CaMG of pigs with colitis [50]. A drop in the populations of ovarian perikarya possessing DβH, but not GAL, also took place in the porcine CaMG after long-term exposure to E_2_ [42] and T [43]. Contrary to the current study, a drop in the TH, NPY and VIP expression in the colon-projecting neuronal cells in CaMG-led colitis [50] and a reduction in the numbers of ovarian perikarya with positive staining for DβH and NPY and immunoreactive to DβH, but not to SOM, was revealed in the CaMG after E_2_ [42] and T [43] administration. The alterations in neurochemical characteristics of the CaMG neurons in gilts after intrauterine *E. coli* injection may be connected with the hormonal state and inflammatory factors having pro-/anti-inflammatory effects. It was reported that receptors for steroid hormones are present in ganglia neurons supplying the reproductive organs in females [41,42,43,51]. The localization of multiple receptors for PGE_2_ [52], LTs [46] and TNF-α [47] was found in the DRG neurons. Although co-expression of steroid hormones and inflammatory mediators’ receptors with particular neurotransmitters in the sympathetic ganglia perikarya have not yet been specified, it cannot be excluded that the differences found depend on the neurochemical characteristics of neurons and their size. This may result from varied density and cellular distribution of receptors for steroid hormones and inflammatory factors in the examined populations of neurons. E_2_ acting by particular ERs-type in cellular line PC12, elevated or reduced transcriptional activity of the TH gene—catecholamine synthesizing enzyme [53]. Moreover, increased synthesis of SP and calcitonin gene-related peptide in rat uterine cervix-related DRG neurons is under the influence of the E_2_-ERs system [54]. Further experiments should be carried out to recognize the co-localization of receptors for steroids and inflammatory mediators with particular neurotransmitters in the CaMG uterine perikarya as well as receptor bases of these substances’ influence on neurochemical characteristics. 

In the *E. coli* group, as found in the current experiment, there was a rise in the number of the CaMG uterine perikarya, which expressed NA, NPY, GAL or VIP, but not SOM. In contrast, a reduction in the populations of noradrenergic and non-noradrenergic perikarya occurred. It is proposed that these alterations may lead to disruptions in a diversity of uterine-sympathetic and non-sympathetic activities. It was revealed that NA exerts an effect on the contractility in healthy [18,19] and inflamed [11] porcine uterus. Similarly, under physiological conditions, NPY, GAL and VIP regulate this uterine activity [22,25,26]. Thus, it is possible that the above-mentioned neurotransmitters may be involved in the elimination of the exudate from the uterus with inflammation. Additionally, NA is able to affect uterine PGs synthesis [20,21], while NPY modulates the uterine blood supply [23,24]. Neuroprotective GAL function in the inflamed uterus may also be possible [55]. Moreover, this study revealed that the inflammatory uterine process led to changes in the uterine small and large perikaryal cell populations, which were partially dependent on their neurochemical features. This may suggest a different action of small and large uterine perikarya on the function of an inflamed uterus. 

## 5. Conclusions

It was found that *E. coli*-induced inflammatory state of the uterus in the pig causes changes in spatial and neurochemical organization patterns of the CaMG neurons innervating the uterus. Alterations in the noradrenergic and non-noradrenergic uterine neuronal populations suggest that inflammation of the uterus may affect the function(s) of this organ by acting on the CaMG uterine neurons. However, further studies are needed to identify the mechanism of the effect of inflammation on uterine innervation. These findings confirm the use of the pig as an appropriate model for the research of the inflammatory states of the reproductive system as well as the importance of the inflammatory process as modulators of neuronal plasticity. Finally, the alterations in the expression of neurotransmitters in the CaMG uterine neurons during uterine inflammation can be used to develop neurotransmitter analogues that restore normal uterine function. 

## Figures and Tables

**Figure 1 animals-10-00891-f001:**
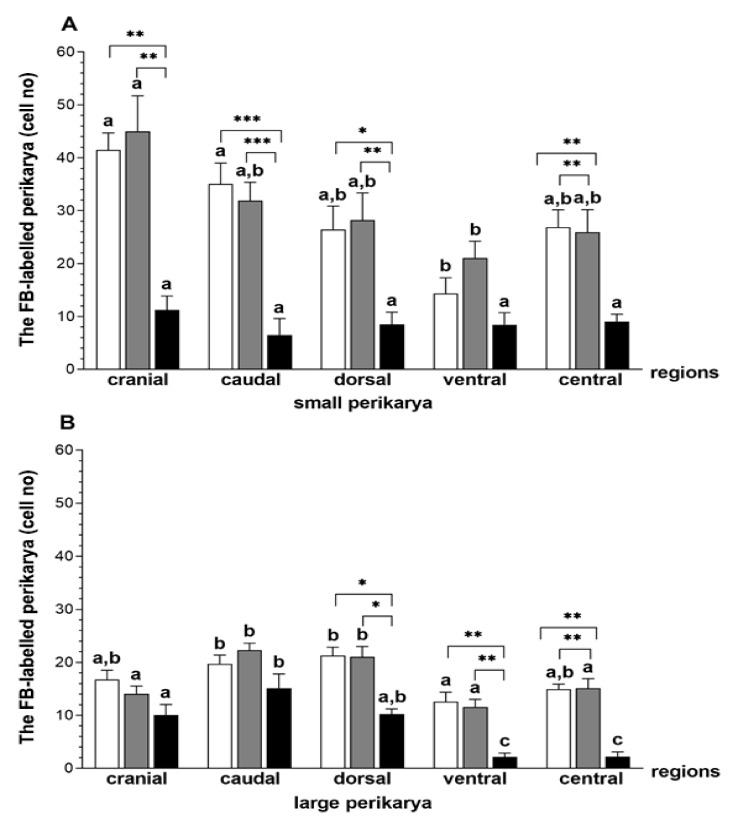
Small (**A**) and large (**B**) perikaryal cell count (mean ± standard error of the mean (SEM)) in the different regions of the caudal mesenteric ganglion (CaMG) projecting to the uterus of gilts from the control (CON) (white bars), saline (SAL) (grey bars) and *E. coli* (black bars) groups. Different letters (a, b, c) show differences (*p* < 0.05–0.001) among the particular regions within the CON, SAL and *E. coli* groups; * *p* < 0.05, ** *p* < 0.01 and *** *p* < 0.001 show differences between all groups in the same ganglion region.

**Figure 2 animals-10-00891-f002:**
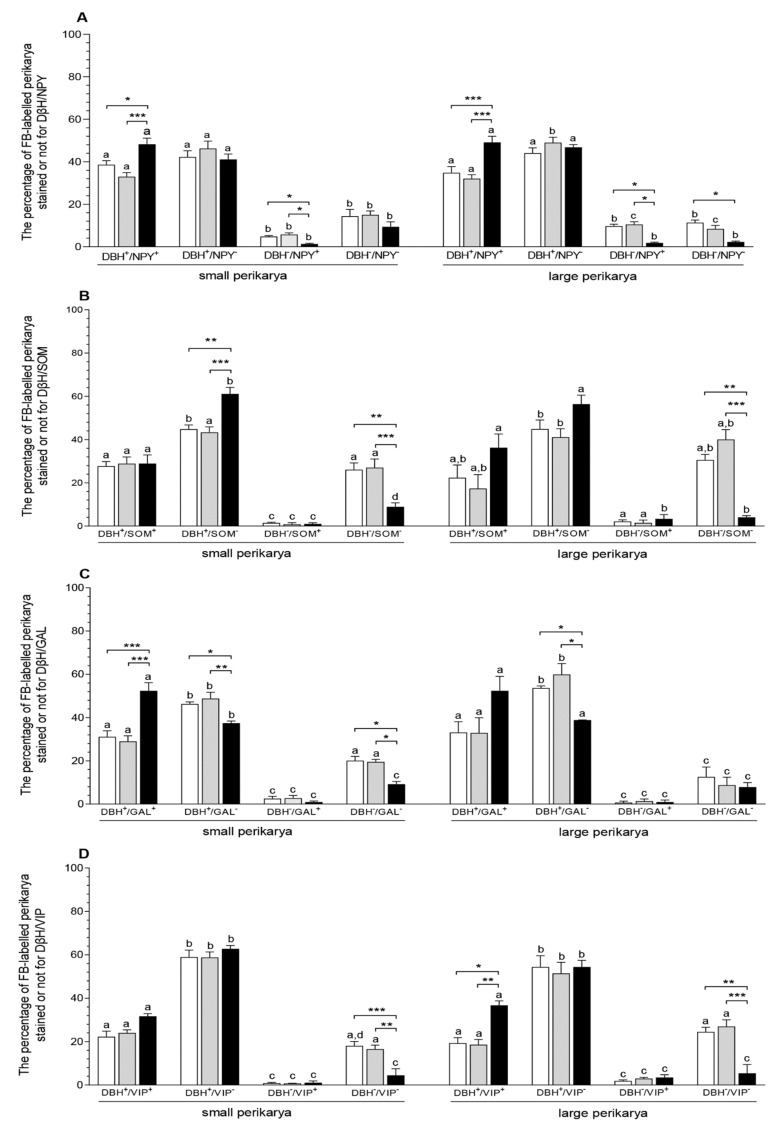
The populations (expressed as percentages, mean ± SEM) of small and large uterine perikarya expressing dopamine-β-hydroxylase (DβH) and/or neuropeptide Y (NPY) (**A**), DβH and/or somatostatin (SOM) (**B**), DβH and/or galanin (GAL) (**C**) and DβH and/or vasoactive intestinal polypeptide (VIP) (**D**) as well as those without these substances in the CaMG of gilts from the CON (white bars), SAL (grey bars) and *E. coli* (black bars) groups. Data are expressed as percentages of the total population of small or large uterine perikarya stained for two substances in each group, accepted as 100%. Different letters (a, b, c, d) show differences (*p* < 0.01, *p* < 0.001) among the particular populations of uterine perikarya within the CON, SAL and *E. coli* groups; * *p* < 0.05, ** *p* < 0.01 and *** *p* < 0.001 show differences between all groups for the same population of uterine perikarya.

**Figure 3 animals-10-00891-f003:**
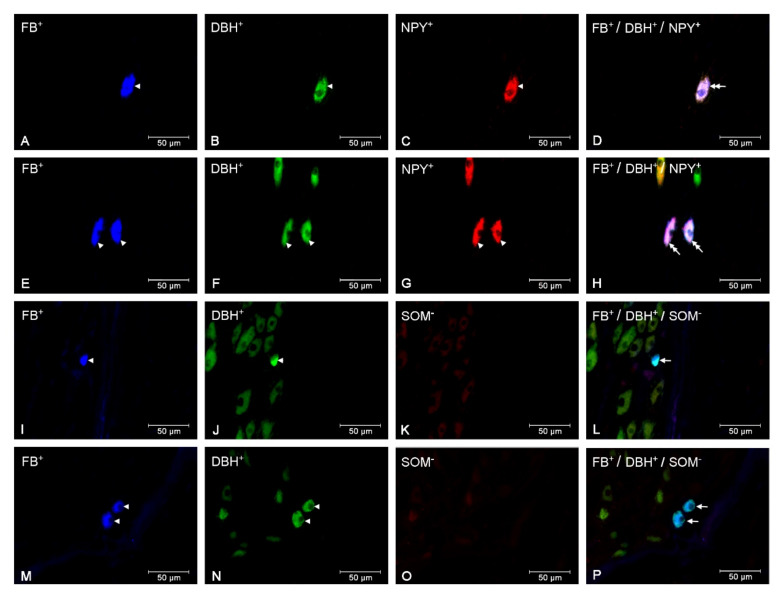
Micrographs demonstrating the presence of DβH (B, F, J, N), NPY (C, G) and SOM (K, O) in the CaMG uterine perikarya of gilts from the CON (**A**–**D**), SAL (**I**–**L**) and *E. coli* (**E**–**H**, **M**–**P**) groups. The arrowhead indicates: Fast Blue (FB)-positive perikaryon, perikaryon presenting positive staining for DβH and NPY and perikaryon presenting positive staining for DβH. The double arrow indicates FB-positive uterine perikaryon expressing DβH and NPY. The arrow indicates FB-positive uterine perikaryon expressing DβH. The photographs (D, H, L, P) were made by digital superimposition of three color channels: FB-positive (blue), DβH-positive (green) and NPY- or SOM-positive (red). One large uterine perikaryon DβH and NPY immunoreactive is visible in the gilt of the CON group (**A**–**D**). In the CaMG of *E. coli* group, an elevation in the population of large perikarya containing these substances is observed (**E**–**H**). One small perikaryon expressing DβH, but not SOM, is present in the ganglion of the SAL group (**I**–**L**). In the *E. coli* group, two small perikarya expressing DβH, but not SOM, are observed in the CaMG (**M**–**P**).

**Figure 4 animals-10-00891-f004:**
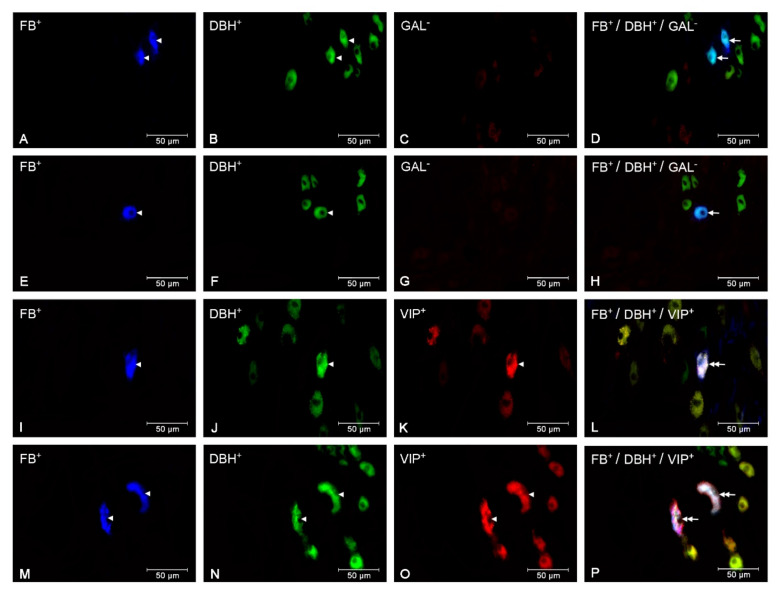
Micrographs demonstrating the presence DβH (B, F, J, N), GAL (C, G) and VIP (K, O) in the CaMG uterine perikarya of gilts from the CON (**A**–**D**), SAL (**I**–**L**) and *E. coli* (**E**–**H**, **M**–**P**) groups. The arrowhead indicates: FB-positive perikaryon, perikaryon presenting positive staining for DβH and perikaryon presenting positive staining for DβH and VIP. The arrow indicates FB-positive uterine perikaryon expressing DβH. The double arrow indicates FB-positive uterine perikaryon expressing DβH and VIP. The photographs (D, H, L P) were made by digital superimposition of three color channels: FB-positive (blue), DβH-positive (green) and GAL- or VIP-positive (red). In the CaMG from the CON group, two small uterine perikarya containing DβH but not GAL are visible (**A**–**D**). Following intrauterine bacterial injection, a drop in the number of small perikarya expressing DβH but immunonegative to GAL is present (**E**–**H**). In the CaMG of gilt from the SAL group, one large perikaryon reactive to DβH and VIP is visible (**I**–**L**). In the gilt of the *E. coli* group, an augmentation in the number of large perikarya occurs (**M**–**P**).

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
