# Peer review of "Endometritis Changes the Neurochemical Characteristics of the Caudal Mesenteric Ganglion Neurons Supplying the Gilt Uterus"

_animals, 2020, doi:10.3390/ani10050891_

Round 1

Reviewer 1 Report

This manuscript attempts to characterize if induction of endometritis in pigs alters the neurochemical characteristics of the sympathetic neurons innervating the uterus.  To this end, the authors injected the neuronal tracer Fast Blue into the uterine horn to label uterus-projecting neurons in the caudal mesenteric ganglia.  After waiting the requisite number of days for the cell bodies to be labeled with dye, the authors injected the uterine horn with saline or E. Coli bacteria to induce inflammation (with no injection also serving as a control).  After 8 days, the animals were euthanized and the mesenteric ganglia removed, sectioned and stained for various transmitters, such as NPY, VIP, galanin, and somatostatin, as well as one of the enzymes responsible for synthesizing norepinephrine.  The authors demonstrate that the total number of Fast Blue positive neurons in the mesenteric ganglia decreases in the pigs with inflammation.  They also describe changes in the transmitters expressed in these neurons, with an increase in DBH/NPY and DBH/GAL double-labeled neurons and decreases in NPY, SOM and GAL single-labeled neurons.  For the most part, the research is well described, and I find no fault with their conclusions.  I have only a few comments/concerns, outlined below: 

Major Concerns:

  1. The English grammar throughout the text should be improved.
  2. The article mentions 4 supplementary figures that are not present. The figure legends supplied appear to be slightly different versions of the figure legends supplied for the figures in the paper.
  3. The statistical markings in Figures 1 and 2 are very confusing; I have no idea what is being compared with the a,b,c… labels.
  4. Moreover, does an experiment number of 3-4 have sufficient power to show statistical significance for some of these differences?
  5. I would have preferred to see the breakdown of transmitters expressed as percentages, rather than (or in addition to) the raw numbers of positive cells.

Author Response

All text improvements of our manuscript have been done in blue font.

Responses to the Review 1

Comments and Suggestions for Authors

This manuscript attempts to characterize if induction of endometritis in pigs alters the neurochemical characteristics of the sympathetic neurons innervating the uterus.  To this end, the authors injected the neuronal tracer Fast Blue into the uterine horn to label uterus-projecting neurons in the caudal mesenteric ganglia.  After waiting the requisite number of days for the cell bodies to be labeled with dye, the authors injected the uterine horn with saline or E. Coli bacteria to induce inflammation (with no injection also serving as a control).  After 8 days, the animals were euthanized and the mesenteric ganglia removed, sectioned and stained for various transmitters, such as NPY, VIP, galanin, and somatostatin, as well as one of the enzymes responsible for synthesizing norepinephrine.  The authors demonstrate that the total number of Fast Blue positive neurons in the mesenteric ganglia decreases in the pigs with inflammation.  They also describe changes in the transmitters expressed in these neurons, with an increase in DBH/NPY and DBH/GAL double-labeled neurons and decreases in NPY, SOM and GAL single-labeled neurons.  For the most part, the research is well described, and I find no fault with their conclusions.  I have only a few comments/concerns, outlined below:

Major Concerns:

  • The English grammar throughout the text should be improved.

English has been corrected by native translator.

  • The article mentions 4 supplementary figures that are not present. The figure legends supplied appear to be slightly different versions of the figure legends supplied for the figures in the paper.

The manuscript does not have supplementary materials. Therefore, Figure legends have been removed.

  • The statistical markings in Figures 1 and 2 are very confusing; I have no idea what is being compared with the a,b,c… labels.

The statistical markings in the Figures 1 and 2 have been corrected (lines 246-248 and 302-304).

  • Moreover, does an experiment number of 3-4 have sufficient power to show statistical significance for some of these differences?

Yes, the number of 3-4 gilts per group has sufficient power to indicate statistical significance between groups. Such numbers of gilts in experimental groups are enough in neuro-immunofluorescence studies conducted on these animals.

  • I would have preferred to see the breakdown of transmitters expressed as percentages, rather than (or in addition to) the raw numbers of positive cells.

On the Figure 2A-D, the numbers of uterine perikarya expressing or not studied substances have been changes into percentages. Statistical analysis of percentage values (mean ±SEM) has been performed.

Reviewer 2 Report

ANIMALS – 801390

 Title: Endometritis Changes the Neurochemical Characteristics of the Caudal Mesenteric Ganglion Neurons Supplying the Gilt Uterus

General Comments:

The authors meant to evaluate the neurochemical characteristics of the caudal mesenteric ganglion neurons supplying the gilt uterus, in endometritis induced condition, by injecting E. coli in the gilt’s uterus. This work relies solely on the immunohistochemistry analysis of caudal mesenteric ganglion neurons. However, in the discussion, the authors refer to the importance of steroid hormones on perikarya cell count in caudal mesenteric ganglion neurons, in order to explain their results. This work would greatly improve if plasma steroid hormone concentrations from those gilts were determined. If that is not feasible, evaluation of steroid receptor expression in caudal mesenteric ganglions could be an option. This way, discussion of the findings is mostly speculation.

The manuscript should be entirely revised. English is poor, and the writing should comply with scientific writing demands. Results and Discussion sections should be rewritten. Figures legends should be improved. Therefore, I would suggest the manuscript to be considered for publication in Animals only after those major issues are thoroughly addressed.

Specific comments

Simple summary:

Line 15: It should read: Most of uterus-projecting neurons are localized in the caudal mesenteric ganglion.

L18: It should read: …. uterine inflammation in pig led to a reduction in the total set of uterine neurons in the mesenteric ganglion, and in the sets of these cells in the dorsal and central areas of this ganglion.

When you refer to “sets” do you mean “groups”?

L21-23- “In the caudal mesenteric ganglion of bacteria-injected gilts, the populations of uterine neurons immunoreactive to dopamine-β-hydroxylase (enzyme participating in noradrenaline synthesis) decreased, while galanin and dopamine-β-hydroxylase were immunonegative , but immunoreactive to neuropeptide Y immunostaining.” This is not clear to me. Was dopamine-β-hydroxylase decreased or absent? Please, clarify this.

L23-25- You say that “ In these gilts the neuronal populations increased, which besides dopamine-β-hydroxylase expressed also neuropeptide Y, galanin and vasoactive intestinal peptide.” This contrasts with what you have said in the previous sentence. Didn´t you mention that galanin and dopamine-β-hydroxylase were immunonegative? If so, how can you say that?

L31- It should read: into each uterine horn~

L 38 It should read: After bacterial treatment, an increase in the numbers of…

L39 - CaMG uterus-supplying neurons, which may

Introduction

L47- It should read:  The uterine inflammatory process is a very common condition in domestic animals, which leads to economic losses.

L53-54- It should read: In turn, in more advanced cases of inflammation, an inflammatory exudate (mucopurulent) is present in the uterine cavity.

L57 - - It should read: ….[11], leukotrienes (LT)B4 and LTC4 [12], as well as nitric oxide [13] are markedly increased.

L58 - It should read:  The uterus in pigs is innervated by nerves from either the autonomic or sensory part of the peripheral nervous system.

L59-60 - It should read:  In the pig, most of uterus-projecting neurons (further referred as uterine neurons), indicated by fluorescent retrograde neuronal tracer Fast Blue (FB), exist in the caudal mesenteric ganglion (CaMG), where they form the “uterine region”.

L65- It should read:  Moreover, in the sow, these neurotransmitters are present in the nerve fibers in the area of CaMG [14,15] and in the uterus, under luminal epithelium, near the blood vessels, glands and myocytes of the myometrium [16,17].

Do you mean endometrial glands? It should be referred.

L68- Please spell out β-ARs before you mention the β adrenergic receptors for the first time in the text.

L70 – Please, spell out Y1Rs

L71 – Please, bear in mind that the word “data” is plural, while “datum” is singular. So, you should write the sentence accordingly:

Data are scarce on the relationship between uterine inflammation and morphology, as

 well as on the immunochemical characteristics of neurons projecting to the uterus.

L76 - up-regulated the uterine substance P (SP)-positive perikarya numbers…

L77 – Due to its embryological, anatomical, and physiological similarities to humans, the pig is a valuable species in biomedical research, also for the study of uterine function.

L78- From one published study conducted on pigs, describing the influence of uterine inflammation on the innervation of the uterus, we have learnt that in gilt Escherichia coli (E. coli)-induced inflammatory state changes the morphological and neurochemical characteristics of the DRG neurons supplying the uterus [32].

L82 - In addition, we found a reduction

L84 - state affects the morphology, as well as the chemical coding of the uterine neurons in CaMG. What do you mean y “chemical coding”?

L96 - By using a tester boar, behavioral estrus was estimated.

L97 - In all studied gilts, disruption in reproductive state did not occur:

L100 - an area of about 5 m2) under the following conditions:

L101 - natural light – day…Was the light natural? It is not clear.

L102 –“The animals were fed typically for the species and age”. What do you mean by fed typically? I am sure the animals were not fed “typically” like in the old times, with potatoes and cabbages. Please, be specific. Were the sows fed commercial diets according to their nutritional requirements?

L102-105- After transport, the pigs were randomly assigned into three groups: Escherichia coli- (E. coli group, n=4), saline -treated gilts (SAL group, n=3), and control gilts (CON control, n=4) - subjected to sham-operation (details are below). The study started after a three-day adaptive period. During the experiment, the animals were not treatment. Please, explain what is meant by this last sentence.

L107 - The experimental procedure was previously described [32].

L108 -… On day 17 of the first studied estrous cycle (day 0 of the study), before surgery, the gilts were premedicated with atropine (0.05mg/kg, administered intramuscularly /i.v./; Atropinum sulf. WZF, Warszawskie ZakÅ‚ady,, Farmaceutyczne Polfa S.A., Poland), azaperone (2 mg/kg BW, administered intramuscularly Stresnil,

 Janssen Pharmaceutica, Beerse, Belgium), and ketamine hydrochloride (10 mg/kg BW, administeredi.v.; Ketamina, Biowet, PuÅ‚awy, Poland). General anesthesia was reached with ketamine

L114- When you say that the uterine horns were removed do you mean you did a hysterectomy? I don´t think that is what you meant. Maybe, you should say the uterine horns were injected with FB…

L117 - uterine horn in paracervical, middle and paraoviductal portions.

L118 - the necessary period for FB to achieve the external sources…

L124 … anaesthetized (as explained above).

L124 - In gilts, after laparotomy was done, either 50 ml of E. coli suspension …

L143 - Sections with FB-positive perikarya were used…

L145 - rinsed in phosphate buffer

L160 - To control for immunofluorescence specificity, standard tests…

L169 – “Perikaryon profiles with a visible nucleus were only scored to avoid double counting.”

It is not clear if you counted the perikarya or not. In the aims of the study you indicated that you examined the (1) total number, size and localization of uterine perikarya, and (2)

 the uterine perikarya numbers containing DβH and/or NPY, SOM, GAL, VIP in the gilt CaMG. 

 Please, clarify this. In the Figures you mention number of FB labelled perikaryal. Is that the mean cell number? How was this count performed? With respect to all the cells

L171 -174 – please, rephrase the sentence. Too many times “constitutes”…

L181- By adding the small and large perikaryal cell count from all areas of the CaMG, the total population of perikarya for each studied group was determined.

L195 - After E. coli treatment the total number of FB-positive perikarya (small and large perikarya combined) was lower (P<0.01) as compared to the CON and SAL groups (70±24.2 vs 228±45.3, 196245±35.1, respectively).

L198- In the E. coli group, a reduction (P<0.01) in the number of small perikarya was

 found in relation to other groups (45.5±14.3 vs. 137.9±18.4, 146±21.4, respectively). Similar findings (P<0.05) concerned the large perikarya (30.8±6.5 vs. 95±25.1, 102±16.8, respectively).  Please, specify which groups you refer to.

L204.205 – This sentence should be deleted. In the previous sentences that information is given with respect to all treatment groups and perikaryal types.

L 207 - Small (A) and large (B) perikarya cell count (mean ±SEM) in CaMG from gilt uterus….

L208 – Do you mean that different letters (a, b, c) show significant differences? Please, clarify this.

Figure 1. In the YY axis, I would advise you to write “ FB-labelled perikarya (cell no)

L214-228 – Please, write this whole section in a clearer manner. You should be scientifically precise and not saying “bacteria injected gilts”, as an example. As far as I understand, you did not inject the gilts with bacteria, but the uterine horns. You should for instances mention DβH negative cells /NPY positive cells before using the acronyms. What you are evaluating is the expression of those proteins in the neurons of the caudal mesenteric ganglion, right? What is meant by “set of DβH / NPY- large perikarya”? Do you mean a group of large perikarya that did not stain for DβH / NPY ? Was that just an isolated group, or a general finding?

I also think that “coded” is not the appropriate term for that. Maybe, you should refer to those cells as those that “presented positive, or negative staining for such protein”.

Fig. 2 – This is confusing. Why do you have in the YY axis “The number of FB- labelled perikaryal” if what you are counting are the cells that stained positive for DβH,  NPY, GAL, and SOM? That should be corrected. Figure legends should be self-explanatory. Do you mean different letters indicate significant differences? Are the perikarya considered “uterine perikarya” or perikarya from caudal mesenteric ganglion neurons?

Fig. 3. It is not clear to me what is meant by “the double arrow that indicates FB-positive uterine perikaryon expressing two studied substances”. In photo A and E, which other substance do you refer to besides FB? These comments should be applied also to the legend of fig. 4.

One large uterine perikaryon DβH and NPY immunoreactive is visible in the gilt of

 CON group (A-D). Just in the caudal mesenteric ganglion of one gilt? This should be clarified.

Discussion

L256- 257 – How can you say that uterine inflammation altered the morphological characteristics of perikarya in CaMG in the pig? All you did was cell count. Did you do any ultrastructural evaluation of those cells? Even though you say that histopathological assessment of pig uteri was previously presented, this is not shown in this article. Besides, one thing is the histopathological assessment of pig´s uterus, another one is the histopathological evaluation of CaMG.

The discussion should be rewritten. The authors make assumptions regarding estrogen and androgen receptors, as well as their blood levels that were not investigated in the present study. The discussion should focus specifically on the findings of the present study. Is the function of large and small perikarya different? Are they different identities or the same cell that undergoes hypertrophy or atrophy? Or is the same cell that, depending on its appearance in the photo, can be measured from different sides and present different sizes? Please indicate the main findings of the study.

Author Response

All text improvements of our manuscript have been done in blue font.

Responses to the Review 2

General Comments:

The authors meant to evaluate the neurochemical characteristics of the caudal mesenteric ganglion neurons supplying the gilt uterus, in endometritis induced condition, by injecting E. coli in the gilt’s uterus. This work relies solely on the immunohistochemistry analysis of caudal mesenteric ganglion neurons. However, in the discussion, the authors refer to the importance of steroid hormones on perikarya cell count in caudal mesenteric ganglion neurons, in order to explain their results. This work would greatly improve if plasma steroid hormone concentrations from those gilts were determined. If that is not feasible, evaluation of steroid receptor expression in caudal mesenteric ganglions could be an option. This way, discussion of the findings is mostly speculation. The manuscript should be entirely revised. English is poor, and the writing should comply with scientific writing demands. Results and Discussion sections should be rewritten. Figures legends should be improved. Therefore, I would suggest the manuscript to be considered for publication in Animals only after those major issues are thoroughly addressed.

Specific comments

Simple summary:

Line 15: It should read: Most of uterus-projecting neurons are localized in the caudal mesenteric ganglion.

It has been corrected according the Reviewer`s suggestion (lines 14 and 15).

L18: It should read: …. uterine inflammation in pig led to a reduction in the total set of uterine neurons in the mesenteric ganglion, and in the sets of these cells in the dorsal and central areas of this ganglion.

It has been corrected according the Reviewer`s suggestion (lines 18-20).

When you refer to “sets” do you mean “groups”?

We referred “sets” to the populations and numbers but not to the groups. “Sets” has been changed into “populations” or “numbers” in the whole manuscript.

L21-23- “In the caudal mesenteric ganglion of bacteria-injected gilts, the populations of uterine neurons immunoreactive to dopamine-β-hydroxylase (enzyme participating in noradrenaline synthesis) decreased, while galanin and dopamine-β-hydroxylase were immunonegative , but immunoreactive to neuropeptide Y immunostaining.” This is not clear to me. Was dopamine-β-hydroxylase decreased or absent? Please, clarify this.

In the corrected version we given: “In the caudal mesenteric ganglion of gilts after intrauterine bacteria injection, the population of uterine neurons presenting positive staining for dopamine-β-hydroxylase (enzyme participating in noradrenaline synthesis) and negative staining for galanin as well as the population of uterine neurons presenting negative staining for dopamine-β-hydroxylase but positive for neuropeptide Y were decreased” (lines 20-24).

L23-25- You say that “ In these gilts the neuronal populations increased, which besides dopamine-β-hydroxylase expressed also neuropeptide Y, galanin and vasoactive intestinal peptide.” This contrasts with what you have said in the previous sentence. Didn´t you mention that galanin and dopamine-β-hydroxylase were immunonegative? If so, how can you say that?

In the corrected version we given: „In these gilts, increased the numbers of uterine neurons which besides dopamine-β-hydroxylase expressed also neuropeptide Y, galanin and vasoactive intestinal peptide” (lines 24-26)..

Explanation of remarks (L21-23, L23-25)

The use of double immunofluorescence method allowed to determine in uterine neurons simultaneously the expression of two different substances (DβH/NPY, DβH/SOM, DβH/GAL, DβH/VIP) in different combination, for example, DβH+/NPY+, DβH+/NPY-, DβH-/NPY+, DβH-/NPY-. (see Figure 2A-D)

We revealed changes (a rise or a decrease) in the numbers of particular populations of perikarya expressing both studied substances or one studied substance or presenting negative staining for both studied substances. Thus, found in this research changes are possible and are not mutually exclusive.

In the Discussion we are trying to explain the cause of changes in the neurochemical features of uterine perikarya in gilts with inflamed uterus.

L31- It should read: into each uterine horn~

It has been corrected according the Reviewer`s suggestion (line 33).

L 38 It should read: After bacterial treatment, an increase in the numbers of…

It has been corrected according the Reviewer`s suggestion (line 40).

L39 - CaMG uterus-supplying neurons, which may

It has been corrected according the Reviewer`s suggestion (line 43).

Introduction

L47- It should read:  The uterine inflammatory process is a very common condition in domestic animals, which leads to economic losses.

It has been corrected according the Reviewer`s suggestion (line 49).

L53-54- It should read: In turn, in more advanced cases of inflammation, an inflammatory exudate (mucopurulent) is present in the uterine cavity.

It has been corrected according the Reviewer`s suggestion (lines 56 and 57).

L57 -  It should read: ….[11], leukotrienes (LT)B4 and LTC4 [12], as well as nitric oxide [13] are markedly increased.

It has been corrected according the Reviewer`s suggestion (line 59).

L58 - It should read:  The uterus in pigs is innervated by nerves from either the autonomic or sensory part of the peripheral nervous system.

It has been corrected according the Reviewer`s suggestion (line 61).

L59-60 - It should read:  In the pig, most of uterus-projecting neurons (further referred as uterine neurons), indicated by fluorescent retrograde neuronal tracer Fast Blue (FB), exist in the caudal mesenteric ganglion (CaMG), where they form the “uterine region”.

It has been corrected according the Reviewer`s suggestion (lines 62-64).

L65- It should read:  Moreover, in the sow, these neurotransmitters are present in the nerve fibers in the area of CaMG [14,15] and in the uterus, under luminal epithelium, near the blood vessels, glands and myocytes of the myometrium [16,17].

Do you mean endometrial glands? It should be referred.

“endometrial” has been added (line 70).

L68- Please spell out β-ARs before you mention the β adrenergic receptors for the first time in the text.

We give full name of “β-ARs” (lines 71 and 72). No abbreviation was given because this name is used only once in the manuscript.

L70 – Please, spell out Y1Rs

We give full name of “Y1Rs” (line 73). No abbreviation was given because this name is used only once in the manuscript.

L71 – Please, bear in mind that the word “data” is plural, while “datum” is singular. So, you should write the sentence accordingly:

It has been corrected (line 77 ).

Data are scarce on the relationship between uterine inflammation and morphology, as  well as on the immunochemical characteristics of neurons projecting to the uterus.

It has been corrected according the Reviewer`s suggestion (lines 77 and 78).

L76 - up-regulated the uterine substance P (SP)-positive perikarya numbers…

It has been corrected according the Reviewer`s suggestion (lines 79 and 80).

L77 – Due to its embryological, anatomical, and physiological similarities to humans, the pig is a valuable species in biomedical research, also for the study of uterine function.

It has been corrected according the Reviewer`s suggestion (lines 80-82).

L78- From one published study conducted on pigs, describing the influence of uterine inflammation on the innervation of the uterus, we have learnt that in gilt Escherichia coli (E. coli)-induced inflammatory state changes the morphological and neurochemical characteristics of the DRG neurons supplying the uterus [32].

It has been corrected (lines 83-85).

L82 - In addition, we found a reduction

It has been corrected according the Reviewer`s suggestion (line 86).

L84 - state affects the morphology, as well as the chemical coding of the uterine neurons in CaMG. What do you mean y “chemical coding”?

“the morphology” has been removed (line 86)

“Chemical coding” has been changed into “neurochemical characteristics” (line 87).  Although “chemical coding” is often used in publications in this field. “Chemical codning”of neurons  means the ability of these cells to synthesize specific neurotransmitters.

“…coded….” was left in the Abstract due to the limited number words required in this part.

L96 - By using a tester boar, behavioral estrus was estimated.

It has been corrected according the Reviewer`s suggestion (lines 100 and 101).

L97 - In all studied gilts, disruption in reproductive state did not occur:

It has been corrected according the Reviewer`s suggestion (line 101).

L100 - an area of about 5 m2) under the following conditions:

It has been corrected according the Reviewer`s suggestion (line 104).

L101 - natural light – day…Was the light natural? It is not clear.

„daylight” has been introduced (line 105).

L102 –“The animals were fed typically for the species and age”. What do you mean by fed typically? I am sure the animals were not fed “typically” like in the old times, with potatoes and cabbages. Please, be specific. Were the sows fed commercial diets according to their nutritional requirements?

It has been explained (lines 105 and 106).

L102-105- After transport, the pigs were randomly assigned into three groups: Escherichia coli- (E. coli group, n=4), saline -treated gilts (SAL group, n=3), and control gilts (CON control, n=4) - subjected to sham-operation (details are below). The study started after a three-day adaptive period. During the experiment, the animals were not treated. Please, explain what is meant by this last sentence.

The animals were not treated in order to not affect the natural course of the inflammatory process (lines 109 and 110).

L107 - The experimental procedure was previously described [32].

It has been corrected according the Reviewer`s suggestion (line 112).

L108 -… On day 17 of the first studied estrous cycle (day 0 of the study), before surgery, the gilts were premedicated with atropine (0.05mg/kg, administered intramuscularly /i.v./; Atropinum sulf. WZF, Warszawskie ZakÅ‚ady,, Farmaceutyczne Polfa S.A., Poland), azaperone (2 mg/kg BW, administered intramuscularly Stresnil, Janssen Pharmaceutica, Beerse, Belgium), and ketamine hydrochloride (10 mg/kg BW, administeredi.v.; Ketamina, Biowet, PuÅ‚awy, Poland). General anesthesia was reached with ketamine

It has been corrected according the Reviewer`s suggestion (lines 112-117).

L114- When you say that the uterine horns were removed do you mean you did a hysterectomy? I don´t think that is what you meant. Maybe, you should say the uterine horns were injected with FB…

It has been changed according the Reviewer`s suggestion (line 119).

L117 - uterine horn in paracervical, middle and paraoviductal portions.

It has been corrected according the Reviewer`s suggestion (line 122).

L118 - the necessary period for FB to achieve the external sources…

It has been corrected according the Reviewer`s suggestion (line 127).

L124 … anaesthetized (as explained above).

It has been corrected according the Reviewer`s suggestion (line 129).

L124 - In gilts, after laparotomy was done, either 50 ml of E. coli suspension …

It has been corrected according the Reviewer`s suggestion (line 129).

L143 - Sections with FB-positive perikarya were used…

It has been corrected according the Reviewer`s suggestion (line 149).

L145 - rinsed in phosphate buffer

It has been corrected according the Reviewer`s suggestion (line 151).

L160 - To control for immunofluorescence specificity, standard tests…

It has been corrected according the Reviewer`s suggestion (line 167).

 L169 – “Perikaryon profiles with a visible nucleus were only scored to avoid double counting.”

It is not clear if you counted the perikarya or not. In the aims of the study you indicated that you examined the (1) total number, size and localization of uterine perikarya, and (2) the uterine perikarya numbers containing DβH and/or NPY, SOM, GAL, VIP in the gilt CaMG.

The sentence has been changed “CaMG uterine perikarya profiles with a visible nucleus were only scored to avoid double counting.” (lines 175 and 176).

The aim of the study has been changed to avoid doubt that uterine perikarya were counted (lines 89-91).

Please, clarify this. In the Figures you mention number of FB labelled perikaryal. Is that the mean cell number? How was this count performed? With respect to all the cells

In our study the CaMG uterine perikarya were counted. Mean numbers of FB-labelled perikarya are mean numbers of uterine neuronal cell bodies.

The exact manner of cell counting is given in Materials and Methods part (lines 171-190).

L171 -174 – please, rephrase the sentence. Too many times “constitutes”…

It has been corrected (lines 178-182).

L181- By adding the small and large perikaryal cell count from all areas of the CaMG, the total population of perikarya for each studied group was determined.

It has been corrected according the Reviewer`s suggestion (lines 178-189).

L195 - After E. coli treatment the total number of FB-positive perikarya (small and large perikarya combined) was lower (P<0.01) as compared to the CON and SAL groups (70±24.2 vs 228±45.3, 196245±35.1, respectively).

L198- In the E. coli group, a reduction (P<0.01) in the number of small perikarya was  found in relation to other groups (45.5±14.3 vs. 137.9±18.4, 146±21.4, respectively). Similar findings (P<0.05) concerned the large perikarya (30.8±6.5 vs. 95±25.1, 102±16.8, respectively).  Please, specify which groups you refer to.

It has been corrected according the Reviewer`s suggestion (lines 204-209).

L204.205 – This sentence should be deleted. In the previous sentences that information is given with respect to all treatment groups and perikaryal types.

It has been removed.

L 207 - Small (A) and large (B) perikarya cell count (mean ±SEM) in CaMG from gilt uterus….

It has been corrected according the Reviewer`s suggestion (lines 246 and 247).

L208 – Do you mean that different letters (a, b, c) show significant differences? Please, clarify this.

Yes, different letters show significant differences. It has been added to the legend (line 247).

Figure 1. In the YY axis, I would advise you to write “ FB-labelled perikarya (cell no)

Axis description was changed  according the Reviewer`s suggestion.

L214-228 – Please, write this whole section in a clearer manner. You should be scientifically precise and not saying “bacteria injected gilts”, as an example. As far as I understand, you did not inject the gilts with bacteria, but the uterine horns. You should for instances mention DβH negative cells /NPY positive cells before using the acronyms. What you are evaluating is the expression of those proteins in the neurons of the caudal mesenteric ganglion, right? What is meant by “set of DβH / NPY- large perikarya”? Do you mean a group of large perikarya that did not stain for DβH / NPY ? Was that just an isolated group, or a general finding? I also think that “coded” is not the appropriate term for that. Maybe, you should refer to those cells as those that “presented positive, or negative staining for such protein”.

The results have been described according the Reviewer`s suggestions (lines 252-272).

Fig. 2 – This is confusing. Why do you have in the YY axis “The number of FB- labelled perikaryal” if what you are counting are the cells that stained positive for DβH,  NPY, GAL, and SOM? That should be corrected. Figure legends should be self-explanatory.

As suggested by the Reviewer 1, on the Figure 2 the populations of uterine perikarya (FB-positive) stained positive for DβH, NPY, SOM, GAL and VIP and uterine perikarya (FB-positive) without staining for these substances are presented as percentages. The descriptions of Y axis has been changed into:

“The percentage of FB-labelled perikarya stained or not for DβH/NPY”

“The percentage of FB-labelled perikarya stained or not for DβH/SOM”

“The percentage of FB-labelled perikarya stained or not for DβH/GAL”

“The percentage of FB-labelled perikarya stained or not for DβH/VIP”

The legend has been also changed (lines 299-305).

Do you mean different letters indicate significant differences?

Yes. It has been added into the legend (lines 303 and 304).

Are the perikarya considered “uterine perikarya” or perikarya from caudal mesenteric ganglion neurons? We have considered the CaMG perikarya projecting their processes to uterus (innervating uterus).

Into the sentence ”In the pig, most of uterus-projecting neurons (further referred as uterine neurons or uterine perikarya), indicated by fluorescent retrograde neuronal tracer Fast Blue (FB), exist in the caudal mesenteric ganglion (CaMG), where they form the “uterine region”. we added phrase “uterine perikarya”, to make it more clear (lines 62-64).

Fig. 3. It is not clear to me what is meant by “the double arrow that indicates FB-positive uterine perikaryon expressing two studied substances”. In photo A and E, which other substance do you refer to besides FB? These comments should be applied also to the legend of fig. 4.

In the Figs 3 and 4 we changed the markers, what is given in the figure legends (lines 308-317 and 320- 329). 

One large uterine perikaryon DβH and NPY immunoreactive is visible in the gilt of  CON group (A-D). Just in the caudal mesenteric ganglion of one gilt? This should be clarified.

On the picture is visible one perikaryon from one gilt. This picture is representative for all gilts from the CON group, according to the statistical analysis.

Discussion

L256- 257 – How can you say that uterine inflammation altered the morphological characteristics of perikarya in CaMG in the pig? All you did was cell count. Did you do any ultrastructural evaluation of those cells? Even though you say that histopathological assessment of pig uteri was previously presented, this is not shown in this article. Besides, one thing is the histopathological assessment of pig´s uterus, another one is the histopathological evaluation of CaMG.

“…morphological characteristics…” has been removed (line 332).

The discussion should be rewritten. The authors make assumptions regarding estrogen and androgen receptors, as well as their blood levels that were not investigated in the present study.

We agree that the Discussion is based on certain assumptions. However, the available data on this area of study are very limited. On the other hand, we try to combine our histochemical observations with its possible functional background.

Obviously, in this study the expression of steroid receptors in uterine perikarya, and peripheral blood levels of estrogen and androgen were not estimated. However, on the base of literature data the steroid influence on the population of uterine perikarya and their neurochemical characteristics can not be excluded. In the corrected version of the manuscript this issue has been limited.

The discussion should focus specifically on the findings of the present study. Is the function of large and small perikarya different? Are they different identities or the same cell that undergoes hypertrophy or atrophy. Or is the same cell that, depending on its appearance in the photo, can be measured from different sides and present different sizes? Please indicate the main findings of the study.

In addition, according to the Reviewer`s suggestion we added considerations about role of the small and large uterine perikarya (lines 424-427), and the changes their populations resulting possibly from neuronal hypertrophy or atrophy (lines 365-371). Used by us the method of measuring the size of uterine perikarya allowed for an exact division into small and large perikarya (lines 183-187).

The main findings of the study have been indicated in the Conclusions part (lines 429 and 430).

Round 2

Reviewer 2 Report

ANIMALS – 801390- R1

 Title: Endometritis Changes the Neurochemical Characteristics of the Caudal Mesenteric Ganglion Neurons Supplying the Gilt Uterus

General Comments:

The authors meant to evaluate the neurochemical characteristics of the caudal mesenteric ganglion neurons supplying the gilt uterus, in endometritis induced condition, by injecting E. coli in the gilt’s uterus. This work relies solely on the immunohistochemistry analysis of caudal mesenteric ganglion neurons. This work was improved, according to the reviewer´s suggestions. However, there are still some minor corrections, regarding English language and sentence construction that should be addressed. Therefore, I would suggest the manuscript be considered for publication in Animals only after those minor issues are thoroughly corrected.

Minor comments

L18-24 – In the caudal mesenteric ganglion of gilts after intrauterine bacterial injection, the population of uterine neurons presenting positive staining for dopamine-β-hydroxylase (an enzyme participating in noradrenaline synthesis) and negative staining for galanin, as well as the population of uterine neurons presenting negative staining for dopamine-β-hydroxylase but positive staining for neuropeptide Y were decreased.

L27.28 - The above changes suggest that inflammation of the gilt uterus may affect the function(s) of this organ by its action on the neurons of the caudal mesenteric ganglion.

L34-35 - The infected gilts presented a severe acute endometritis.

.

L37 - In the CaMG, bacterial injection decreased…Please, write “bacterial injection” instead of “bacteria injection” throughout the text.

L40 - bacterial treatment – please correct it throughout the text.

L89 - on (1) the total number of uterine perikarya and their size and localization, and on (2) the uterine perikaryal cell count

L126  ….innervation of the uterus in pigs)

L186 - By adding the small and large perikaryal cell count from all areas…

L199 - in which 3 to 4 animals were used for neuro-immunofluorescence experiments.

L212- the bacterial-treated gilts

L245- Fig. 1 perikaryal cell count….

L255- intrauterine bacterial infusion

L338 - The current study revealed that after intrauterine infusion of bacteria in the gilt, the total population of uterine perikarya in CaMG was reduced.

L348-349- to induce growth factors production, as well as their antioxidant actions.

L351 . long-term T administration. Does T stand for testosterone?

L356 - pro-inflammatory cytokines (TNF-α, IL-1β), as well as… A comma should be placed before “as well as”).

L361- ARs [42,43], as well as receptors…(A comma should be placed before “as well as”).

L365- In addition, in the DRGs of gilts used in this study

L367-370 - Although in the present experiment the effect of the inflammatory process on the dynamic changes in perikarya size has not been determined, the revealed changes in the numbers of small and large uterine perikarya may result from their atrophy or hypertrophy, although this hypothesis needs confirmation.

L373- In addition, uterine inflammation decreased the numbers…

L 376 - Partly compatible to the current findings is another study, which reported a drop

L378 - ovarian perikarya in these ganglia

L384-388 – In contrast, a decrease was noted in the number of small and large uterine perikarya expressing DβH but not GAL; in the number of small and large uterine perikarya immunonegative to DβH, but immunoreactive to NPY, as well as in the number of uterine perikarya presenting negative staining for the studied substances (except for small uterine perikarya immunonegative to DβH and NPY and large uterine perikarya immunonegative to DβH and GAL)

L404 - the neurochemical characteristics of neurons and their size. This may result from varied density and cellular distribution of receptors for steroid hormones and inflammatory factors in the examined populations of neurons.

L413 – In the E. coli group, as found in the current experiment, there was a rise in the number of the CaMG uterine perikarya, which expressed NA, NPY, GAL or VIP, but not SOM.  In contrast, a reduction in the populations of noradrenergic and non-noradrenergic perikarya occurred.

423-Moreover, this study revealed that the inflammatory uterine process led to changes in the uterine small and large perikaryal cell populations, which were partially dependent on their neurochemical features.

Author Response

All text improvements of our manuscript have been done in green font.

Responses to rev. II

Author's Reply to the Review Report (Reviewer 2)

ANIMALS – 801390- R1

 Title: Endometritis Changes the Neurochemical Characteristics of the Caudal Mesenteric Ganglion Neurons Supplying the Gilt Uterus

General Comments:

The authors meant to evaluate the neurochemical characteristics of the caudal mesenteric ganglion neurons supplying the gilt uterus, in endometritis induced condition, by injecting E. coli in the gilt’s uterus. This work relies solely on the immunohistochemistry analysis of caudal mesenteric ganglion neurons. This work was improved, according to the reviewer´s suggestions. However, there are still some minor corrections, regarding English language and sentence construction that should be addressed. Therefore, I would suggest the manuscript be considered for publication in Animals only after those minor issues are thoroughly corrected.

Minor comments

L18-24 – In the caudal mesenteric ganglion of gilts after intrauterine bacterial injection, the population of uterine neurons presenting positive staining for dopamine-β-hydroxylase (an enzyme participating in noradrenaline synthesis) and negative staining for galanin, as well as the population of uterine neurons presenting negative staining for dopamine-β-hydroxylase but positive staining for neuropeptide Y were decreased.

It has been corrected according the Reviewer`s suggestion (lines 21-25).

L27.28 - The above changes suggest that inflammation of the gilt uterus may affect the function(s) of this organ by its action on the neurons of the caudal mesenteric ganglion.

It has been corrected according the Reviewer`s suggestion (lines 27-29).

 L34-35 - The infected gilts presented a severe acute endometritis.

It has been corrected according the Reviewer`s suggestion (lines 36 and 37).

L37 - In the CaMG, bacterial injection decreased…Please, write “bacterial injection” instead of “bacteria injection” throughout the text.

It has been corrected according the Reviewer`s suggestion.

L40 - bacterial treatment – please correct it throughout the text.

It has been corrected according the Reviewer`s suggestion.

L89 - on (1) the total number of uterine perikarya and their size and localization, and on (2) the uterine perikaryal cell count

It has been corrected according the Reviewer`s suggestion (lines 91 and 92).

L126  ….innervation of the uterus in pigs)

It has been corrected according the Reviewer`s suggestion (line 129).

 L186 - By adding the small and large perikaryal cell count from all areas…

It has been corrected according the Reviewer`s suggestion (line 188).

 L199 - in which 3 to 4 animals were used for neuro-immunofluorescence experiments.

It has been corrected according the Reviewer`s suggestion (lines 201 and 202).

L212- the bacterial-treated gilts

It has been corrected according the Reviewer`s suggestion (line 214).

 L245- Fig. 1 perikaryal cell count….

It has been corrected according the Reviewer`s suggestion (line 247).

 L255- intrauterine bacterial infusion

It has been corrected according the Reviewer`s suggestion (line 257).

L338 - The current study revealed that after intrauterine infusion of bacteria in the gilt, the total population of uterine perikarya in CaMG was reduced.

It has been corrected according the Reviewer`s suggestion (lines 335 and 336).

L348-349- to induce growth factors production, as well as their antioxidant actions.

It has been corrected according the Reviewer`s suggestion (lines 345 and 346).

L351 . long-term T administration. Does T stand for testosterone?

Full name has been given (line  349).

L356 - pro-inflammatory cytokines (TNF-α, IL-1β), as well as… A comma should be placed before “as well as”).

It has been corrected according the Reviewer`s suggestion (line 354).

L361- ARs [42,43], as well as receptors…(A comma should be placed before “as well as”).

It has been corrected according the Reviewer`s suggestion (line 358).

L365- In addition, in the DRGs of gilts used in this study

It has been corrected according the Reviewer`s suggestion (line 363).

L367-370 - Although in the present experiment the effect of the inflammatory process on the dynamic changes in perikarya size has not been determined, the revealed changes in the numbers of small and large uterine perikarya may result from their atrophy or hypertrophy, although this hypothesis needs confirmation.

It has been corrected according the Reviewer`s suggestion (lines 365-368).

 L373- In addition, uterine inflammation decreased the numbers…

It has been corrected according the Reviewer`s suggestion (line 371).

 L 376 - Partly compatible to the current findings is another study, which reported a drop

It has been corrected according the Reviewer`s suggestion (lines 374 and 375).

 L378 - ovarian perikarya in these ganglia

It has been corrected according the Reviewer`s suggestion (line 376).

L384-388 – In contrast, a decrease was noted in the number of small and large uterine perikarya expressing DβH but not GAL; in the number of small and large uterine perikarya immunonegative to DβH, but immunoreactive to NPY, as well as in the number of uterine perikarya presenting negative staining for the studied substances (except for small uterine perikarya immunonegative to DβH and NPY and large uterine perikarya immunonegative to DβH and GAL)

It has been corrected according the Reviewer`s suggestion (lines 381-386).

L404 - the neurochemical characteristics of neurons and their size. This may result from varied density and cellular distribution of receptors for steroid hormones and inflammatory factors in the examined populations of neurons.

It has been corrected according the Reviewer`s suggestion (lines 401-403).

 L413 – In the E. coli group, as found in the current experiment, there was a rise in the number of the CaMG uterine perikarya, which expressed NA, NPY, GAL or VIP, but not SOM.  In contrast, a reduction in the populations of noradrenergic and non-noradrenergic perikarya occurred.

It has been corrected according the Reviewer`s suggestion (lines 410-412).

L423-Moreover, this study revealed that the inflammatory uterine process led to changes in the uterine small and large perikaryal cell populations, which were partially dependent on their neurochemical features.

It has been corrected according the Reviewer`s suggestion (lines 420-422).

This manuscript is a resubmission of an earlier submission. The following is a list of the peer review reports and author responses from that submission.